# Psychometric properties of upper limb kinematics during functional tasks in children and adolescents with dyskinetic cerebral palsy

Inti Vanmechelen[1]*, Saranda Bekteshi[1], Marco Konings[1], Hilde Feys[2], Kaat Desloovere[3], Jean-Marie Aerts[4], Elegast Monbaliu[1]

1 Department of Rehabilitation Sciences, KU Leuven Bruges, Bruges, Belgium, 2 Department of Rehabilitation Sciences, KU Leuven, Leuven, Belgium, 3 Department of Rehabilitation Sciences, KU Leuven, Pellenberg, Belgium, 4 Department of Biosystems, Division of Animal and Human Health Engineering, Measure, Model and Manage Bioresponse (M3-BIORES), KU Leuven, Leuven, Belgium

* Inti.vanmechelen@kuleuven.be

## Abstract

Dyskinetic cerebral palsy (DCP) is characterised by involuntary movements, and the movement patterns of children with DCP have not been extensively studied during upper limb tasks. The aim of this study is to evaluate psychometric properties of upper limb kinematics in participants with DCP and typically developing (TD) participants. In current repeatability and validity study, forty individuals with typical development (n = 20) and DCP (n = 20) performed a reach forward/sideways and a reach and grasp task during motion analysis on two occasions. Joint angles at point of task achievement (PTA) and spatio-temporal parameters were evaluated within-and between-sessions using intra-class correlation coefficients (ICC) and standard error of measurement (SEM). Independent t-tests/Mann-Whitney-U tests were used to compare parameters between groups. Within-session ICC values ranged from 0.45 to 1.0 for all parameters for both groups. Within-session SEM values ranged from 1.1˚ to 11.7˚ for TD participants and from 1.9˚ to 13.0˚ for participants with DCP. Eight within-session repetitions resulted in the smallest change in ICC and SEM values for both groups. Within-session variability was higher for participants with DCP in comparison with the TD group for the majority of the joint angles and spatio-temporal parameters. Intrinsic variability over time was small for all angles and spatio-temporal parameters, whereas extrinsic variability was higher for elbow and scapula angles. Between-group differences revealed lower shoulder adduction and higher elbow flexion, pronation and wrist flexion, as well as higher trajectory deviation and a lower maximal velocity for participants with DCP. This is the first study to assess the psychometric properties of upper limb kinematics in children and adolescents with DCP, showing that children with DCP show higher variability during task execution, requiring a minimum of eight repetitions. However, their variable movement pattern can be reliably captured within-and between-sessions, confirming the potential of three-dimensional motion analysis for assessment of rehabilitation interventions in DCP.

**Data Availability Statement:** All relevant data are within the paper and its Supporting information files.

**Funding:** IV is funded by Fonds Wetenschappelijk onderzoek Vlaanderen (FWO), grant number 65831. https://www.fwo.be/. The funder did not play a role in the study design, data collection and analysis, decision to publish, or preparation of the manuscript.

**Competing interests:** The authors have declared that no competing interests exist.

## Introduction

Cerebral palsy (CP) is 'a group of permanent disorders of the development of movement and posture, attributed to non-progressive disturbances that occurred in the developing fetal or infant brain.' [1]. Based on the predominant movement disorders, patients with CP are classified into spastic, dyskinetic, ataxic, or mixed forms of CP [2]. Dyskinetic CP (DCP) is the second most common type of CP with a prevalence of 14.4% and is characterized by a combination of disturbed movement control and a varying muscle tone [2, 3]. Since 50% of the patients with DCP are wheelchair-bound, an optimal function of the upper limbs, ensuring wheelchair propulsion or cutlery handling, is of paramount importance to maintain an independent lifestyle [4]. DCP includes two major movement disorders: dystonia and choreoathetosis [5]. Dystonia is defined as 'a movement disorder characterized by sustained or intermittent muscle contractions causing abnormal, often repetitive movements, postures or both' [6]. Choreoathetosis is characterized by hyperkinesia and muscle tone fluctuation, leading to jerky and constantly changing movements [5].

Dystonia and choreoathetosis in patients with DCP is currently evaluated through the use of clinical assessment tools. Several qualitative assessment tools have recently been presented to evaluate dystonia [7, 8], of which some specifically for dystonia in CP [8]. Only one assessment scale is currently available to assess both dystonia and choreoathetosis in DCP [9]. However, the ordinal scoring in such qualitative assessments diminishes the score variability and induces a lower responsiveness [10]. Moreover, a score is defined based on video measurements and consensus definitions, making data analyses time-consuming and subjective. Last, these qualitative assessment tools focus on the presence and severity of dystonia and choreoathetosis, but do not yield information on specific movement patterns. The latter is, however, considered crucial for goal-directed training towards improved upper limb performance and to evaluate the effect of treatments.

Over the past years, there have been several attempts to establish objective measurements in the CP population. Gordon et al., [11] attempted to discriminate dystonia and spasticity in the arm where spasticity was expressed as the amount of force necessary to passively extend the elbow joint as measured with a rigidity analyser and dystonia was characterized as the amount of overflow movement in the contralateral arm. However, evaluating the amount of dystonia only by overflow movements of the contralateral arm does not capture the full aspect of dystonia and its action-specific aspect. Sanger et al. [12] demonstrated an increased movement variability and a lack of straight-line trajectories in participants with DCP during outward reaching. While these results indicate the ability to quantitatively measure movement characteristics of the upper limb using position diodes attached to eight points of the body, they do not provide any information regarding joint angles or movement patterns. When focusing on hemiplegic spastic CP, several upper limb protocols have been developed and validated over the past years [13–17]. While all studies presented moderate to good results, the upper limb joints included in the analyses were limited to trunk, shoulder, elbow and wrist angles. The study of Jaspers et al. was the only protocol so far that has additionally presented scapular angles, allowing to investigate the role of the scapula position in upper arm movements [18, 19]. In all abovementioned protocols, analysis of joint kinematics demonstrated significant differences between typically developing (TD) children and children with hemiplegic spastic CP [17, 20–22], most frequently in elbow extension and elbow supination angles. To date, only one study on kinematic analysis of upper limb movements included children with DCP, representing only a small sub-group of the patient cohort [23]. We currently do not know anything about the movement patterns in individuals with DCP as recorded with three-dimensional

motion analysis, which currently prohibits us in using this methodology to evaluate the effect of rehabilitation strategies.

As dyskinetic CP is characterized by involuntary movements, it is expected that their movement patterns will be less consistent compared to TD children or children with spastic CP. In this perspective, we strive towards reliably capturing a pattern that is inherently inconsistent, which may thus require a higher number of repetitions within one session before parameter calculation.

Since novel assessments need to be reliable and valid before they can be transferred to clinical practice, the objective of this study is to evaluate the psychometric properties of upper limb kinematics in children and adolescents with and without DCP. The first goal focuses on repeatability, where the objective is to define the within-session repeatability of joint angles and spatio-temporal parameters and to explore the number of repetitions that are necessary within one session to obtain a representative and robust representation of the movement pattern for participants with and without DCP. The hypothesis is that a higher number of repetitions in comparison with spastic CP is necessary for a robust representation [12]. The second goal focuses on the increased variability in the movement patterns of individuals with DCP. The objective is to assess the variability between TD participants and participants with DCP for this specified number of repetitions. We hypothesize that participants with DCP show higher variability in comparison with their TD peers. The third goal focuses on between-session measures. The objective is to assess between-session repeatability of the joint angles and spatio-temporal parameters, as this is an important first step toward responsiveness of these measures. The hypothesis is that joint angles and spatio-temporal parameters can be reliably captured over time. The fourth goal focuses on validity. The objective is to evaluate discriminative validity of three-dimensional motion measures, defining the differences in upper limb kinematics between children and adolescents with and without DCP. The hypothesis is that the joint angles and spatio-temporal parameters will differ significantly between the TD and DCP group.

## Methods

### Study design

Within-session reliability and repeatability were evaluated using the intra-class correlation coefficient and standard error of measurement on the parameters collected within one session. Between-session repeatability was evaluated by using data of the first and second session, and intrinsic and extrinsic variability were explored. All parameters were compared between the TD individuals and individuals with DCP to evaluate between-group differences. The Cosmin checklist was used for standardisation of reporting of clinimetric properties and we adhered to the SPICES method to ascertain inclusion of all aspects of the methodology [24, 25].

### Participants

Participants were recruited from special education schools across Belgium for children with multiple disabilities, and from the University Hospitals Leuven. Individuals with DCP were eligible to participate if they: were diagnosed with DCP by a paediatric neurologist, were aged between 5–25 years old and were classified as Manual Ability Classification System (MACS) level I-III [26]. Exclusion criteria were: a neurological disorder other than DCP, botulinum toxin injections in the upper limb muscles in the past 6 months and neurological or orthopaedic surgery in the last year before assessment. TD participants were recruited from a peripheral network and eligible to participate if they were aged between 5–25 years. With respect to ethics, all participants and/or their parents provided written consent prior to participation in

accordance with the Declaration of Helsinki. The study was approved by the Ethics committee research UZ / KU Leuven, S-number S62093.

## Study procedures

Every child was evaluated twice on the same day with a minimum of one hour and a maximum of two hours between sessions at the WE-lab for Health, Technology and Management (KU Leuven, campus Bruges) or the Clinical Movement Analysis Laboratory (CMAL, UZ Leuven, Pellenberg) by the same assessors. All participants were asked to perform three upper limb tasks: reaching forward (RF), reaching sideways (RS) and reach and grasp vertical (RGV). RF, RS and RGV were executed at shoulder height (acromion) and reaching distance was determined according to arm length (from acromion to caput metacarpal III). All tasks were performed at self-selected speed with the non-preferred arm (the hemiplegic arm in participants with unilateral DCP and the non-preferred arm in TD participants) and with both arms in participants with bilateral DCP. Start position (the ipsilateral knee) was indicated with an elastic band above the knee. Every task was executed 10 times per trial with a total of three trials for every task. Participants were seated in a chair with adjustable height and a custom-made reaching system was developed to perform the tasks in a standardized way (Fig 1). The reference position was 90˚ flexion in hip and knees and the hands placed on the ipsilateral knee [19]. Seventeen reflective markers were placed over the body in 5 clusters: two cuffs of 4 markers were placed respectively on the upper arm and forearm, one cluster of 3 markers was placed on the hand and two tripods with 3 markers were placed respectively on the trunk and the scapula. Five segments were thus included (trunk, scapula, humerus, forearm, hand) and four joints were considered (scapulothoracic (scapula), humerothoracic (shoulder), elbow, wrist). Anatomical landmarks were palpated according to precise definitions and digitized using a pointer with four linear markers and anatomical coordinate systems and joint rotation sequences were defined according to the ISB-guidelines [19, 27]. Static and dynamic calibrations were subsequently performed for the calculation of anatomical landmarks, during which passive assistance was given where needed. 3D marker tracking was done with 12 infra-red Vicon optical motion capture cameras sampling at 100 Hz and 2 high-definition video cameras, with a typical measurement error of 0.4 mm (Vicon Motion Systems, Oxford Metrics, UK). The currently used protocol has been previously validated in TD participants and participants with hemiplegic spastic CP [18, 19].

## Data analysis

Movement cycles were identified and segmented in Vicon Motion Capture System. One movement cycle was defined from hand on ipsilateral knee to point of task achievement

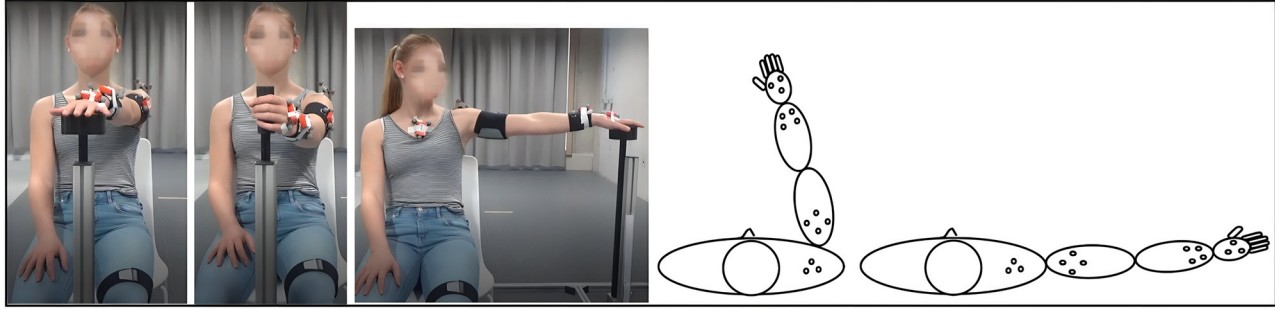

**Fig 1. Functional upper limb tasks: Reach forwards, reach sideways and reach to grasp vertically, with an illustration of the cluster markers on the shoulder, upper arm, wrist and hand for reach forwards and reach sideways.**

(PTA), where PTA is considered the final point of the reaching or reach-and-grasp cycle [19]. The first and last movement cycles were disregarded as they could be influenced by stop and start strategies, resulting in 8 repetitions for each trial, with a total of 24 repetitions for each task. Joint angle at PTA was obtained by selecting the last value of the angular waveform for the joint angles. Subsequently, maximal velocity and trajectory deviation were obtained for each repetition. Trajectory deviation is a dimensionless parameter, but a value of 1 implies a perfect straight line trajectory, whereas the higher the trajectory deviation, the more the movement deviates from a straight line.

To evaluate how many repetitions of a task execution represented a stable movement pattern, an incremental number of repetitions was randomly selected for each task and the change in outcome values was evaluated for both TD participants and participants with DCP.

In case of missing data, the joint angle for which the data is missing was excluded from the subsequent data analysis for this specific participant.

## Statistical analysis

*Within-session repeatability*: *Goal 1*—For both groups, 2, 4, 6, 8 and 10 repetitions from the RF, RGV and RS task were randomly selected for each task for each participant. Subsequently, intra-class correlation coefficient (ICC) values—$ICC_w(2,1)$ based on single measures [28]—and standard error of measurement (SEM) were calculated for each number of repetitions for the joint angle at PTA and the spatio-temporal parameters for each functional task. Values of ICC were interpreted as poor ($<0.50$), moderate ($0.50- < 0.75$), good ($0.75–0.90$), and excellent ($> 0.90$) [29]. SEM calculations were based on the square root of the within-group mean square value of the one-way ANOVA [30].

The change in both the ICC and SEM values for the different repetitions (2, 4, 6, 8 or 10) was expressed in percentage (%) of change in comparison with the highest SEM or ICC value for all number of repetitions. The cut-off value for a stable ICC or SEM value was defined as the difference between incrementing repetitions being less than or equal to 10%. The SEM defining the cut-off value will hereafter be referred to as 'consistency measure', since we assume that this margin of error defines a consistent performance within one session.

*Assessment of variability*: *Goal 2*: To evaluate whether the variability was higher for participants with DCP in comparison with their TD peers, standard deviations for the selected number of repetitions were calculated and compared between groups using an independent t-test/ Mann Whitney-U test depending on the data distribution.

*Between-session repeatability*: *Goal 3*—To evaluate repeatability over time in patients with DCP, it is important to differentiate between internal variability (the difference in consistency only related to the participants' performance) and external sources of variability (e.g. marker placement and palpation differences). To evaluate internal variability, we compared the consistency measure between session 1 and session 2 for each task. To evaluate external variability we compared the mean of the consistency measures of session 1 and session 2 with the between-session standard error for all joint angles and spatio-temporal parameters.

*Between-group comparison*: *Goal 4*—Joint angles at PTA and spatio-temporal parameters averaged over repetitions were assessed for normality and compared between groups with an independent t-test/Mann Whitney-U test. Additionally, absolute differences (the difference between the mean of the TD and DCP group) were compared with the between-session standard error to evaluate for which parameters the absolute difference exceeds the standard error.

The sample size was based on outcome parameters (i.e. joint angles) of a previous validity study comparing spastic CP patients with their TD peers, yielding an effect size of 0.91 [20]. Based on this effect size, a group of 20 DCP and 20 TD individuals was sufficient.

All analyses were performed in SPSS 28.0.0.0 with the significance level set at p<0.05 (SPSS Inc., Chicago, IL).

## Results

### Participants

Twenty participants with DCP (mean age 16y6m, age range 8-25y) were available for a first data collection, 16 of these participants were also evaluated in the second session. For four participants with bilateral DCP, both arms were measured and included as separate data subjects. Since dyskinetic CP is characterized by asymmetry and involuntary movements, we assured that inclusion of the second arm did not distort the results on a group level. Twenty TD participants (mean age 17y1m, age range 9-24y) were available for a first data collection, from which six TD participants were also recorded for a second session. Participant characteristics are summarized in S1 Table. Fourteen participants from the DCP group were right-handed and six participants were left-handed, 17 participants from the TD group were right-handed, three were left-handed. Two participants from the DCP group were unable to perform the reach and grasp vertical task and one participant with DCP did not perform the reach sideways task due to fatigue. For four participants (2 TD; 2 DCP), the values for shoulder rotation and elevation plane during the reach forward and reach and grasp vertical task were incorrect and removed from the analyses. The ICC and SEM values for reach and grasp vertical are thus based on 18 participants with DCP for all angles except for elevation plane and shoulder rotation (16 participants) and 18 TD participants. The ICC and SEM values for reach sideways are based on 19 participants with DCP and 20 TD participants.

### Within-session repeatability

The ICC values for joint angles at PTA ranged from 0.52 to 0.98 for TD participants and from 0.45 to 1.0 for participants with DCP for all tasks (Fig 2). For both groups, there were no changes higher than 10% in ICC value after four repetitions (S2 Table). The SEM values for joint angles at PTA ranged from 1.1˚ to 11.7˚ for TD participants and from 1.9˚ to 13.0˚ for participants with DCP (Fig 3). For the TD group, there were no changes higher than 10% in SEM after six repetitions for RF, after four repetitions for RS and after eight repetitions for RGV. For the DCP group, there were no changes higher than 10% in SEM after eight repetitions for all tasks (S2 Table).

For the spatio-temporal parameters (Fig 4), ICC values for trajectory deviation ranged from 0.72 to 0.85 during RF and from 0.51 to 0.63 during RGV and RS for TD participants. For participants with DCP, ICC values ranged from 0.64 to 0.85 during all functional tasks. ICC's for maximal velocity ranged from 0.71–0.92 for all functional tasks for TD participants. For participants with DCP, ICC's for maximal velocity ranged from 0.52 to 0.73 during RF, and from 0.74–0.86 during RGV and RS. For both groups, there was no increase in ICC value higher than 10% for any of the parameters after six repetitions (S3 Table). For the DCP group, SEM values for trajectory deviation ranged from 0.09 to 0.23 and from 0.15 to 0.18 for maximal velocity. For the TD group, SEM values for trajectory deviation ranged from 0.02 to 0.05 and from 0.07 to 0.15 for maximal velocity. For both groups, there was no increase in ICC value higher than 10% for any of the parameters after six repetitions (S3 Table).

Since the majority of the standard deviations was not normally distributed, groups were compared with the Mann Whitney-U test. The standard deviations for the joint angles were significantly higher for the DCP group in comparison with the TD group during RF for all joint angles except shoulder rotation, scapula pro/retraction and scapular tilting (Fig 5, S4 Table). During RGV, standard deviations were significantly higher for elevation plane,

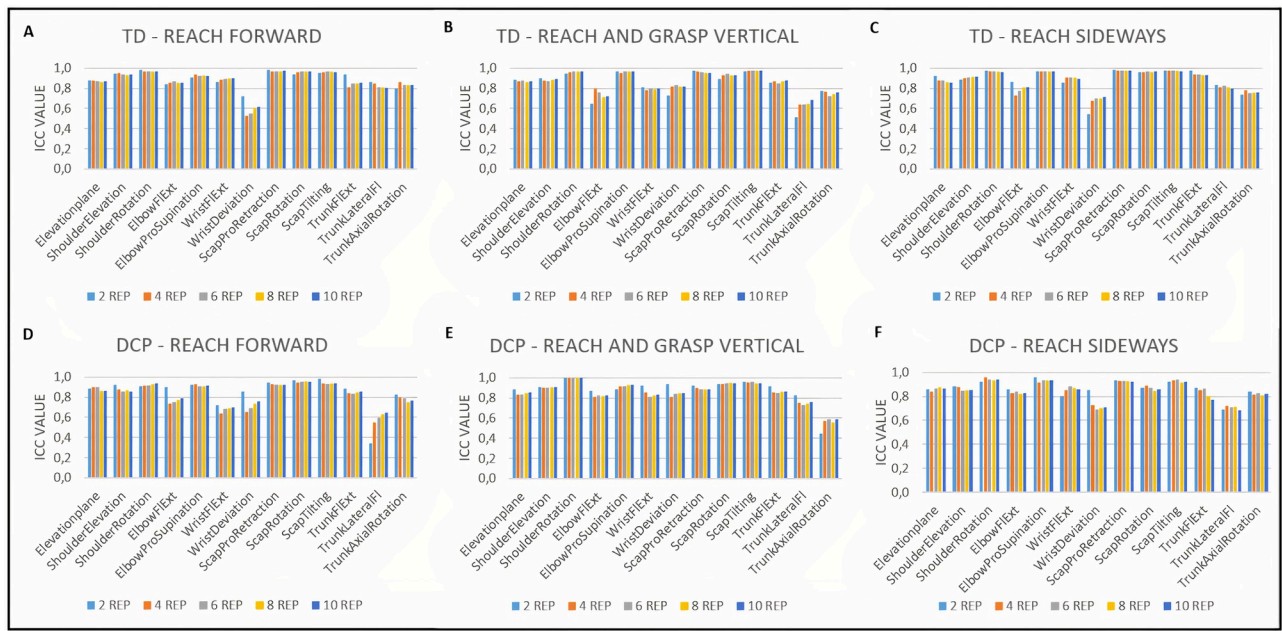

**Fig 2. Within-session intra-class correlation coefficients for joint angles at point of task achievement for TD participants (A,B,C) and participants with dyskinetic cerebral palsy (DCP) (D, E, F).** Fl = flexion; Ext = extension; Scap = scapular; Pro = protraction; REP = repetitions.

shoulder elevation, elbow pro/supination and all scapula and trunk angles and during RS, standard deviations were higher for the DCP group for all angles except elevation plane, shoulder rotation, elbow flexion/extension and pro/supination. For the spatio-temporal parameters, both maximal velocity and trajectory deviation showed higher standard deviations for the DCP group for all tasks.

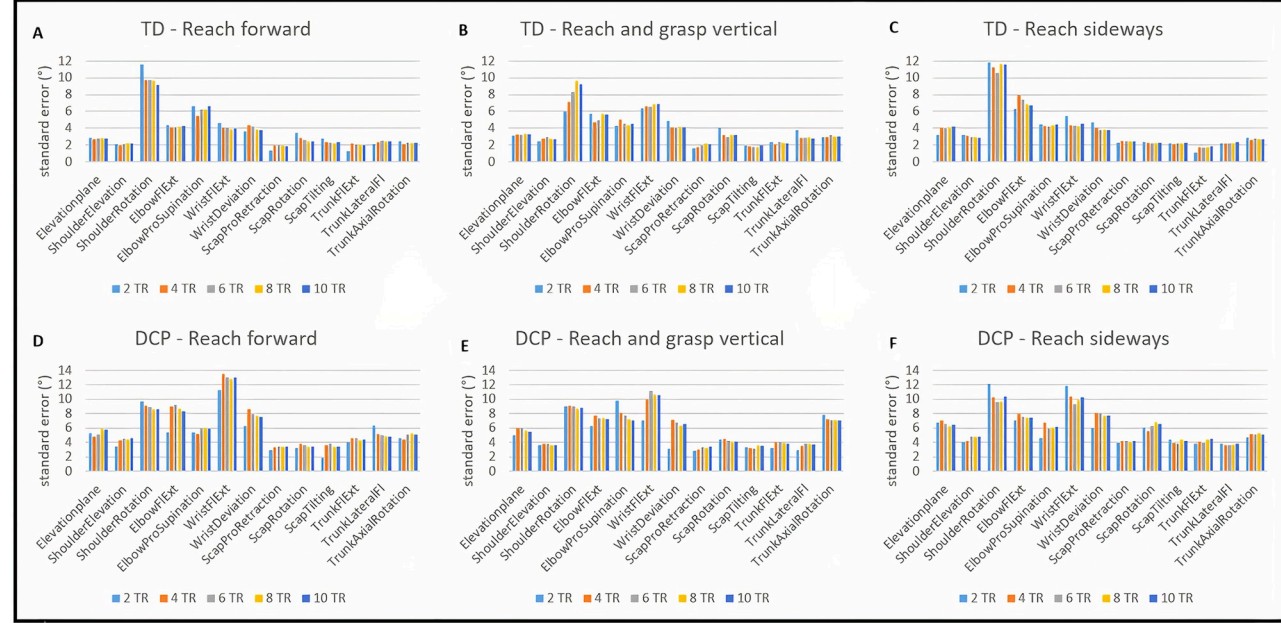

**Fig 3. Within-session standard error of measurement for joint angles at point of task achievement for TD participants (A,B,C) and participants with dyskinetic cerebral palsy (DCP) (D, E, F).** Fl = flexion; Ext = extension; Scap = scapular; Pro = protraction; REP = repetitions.

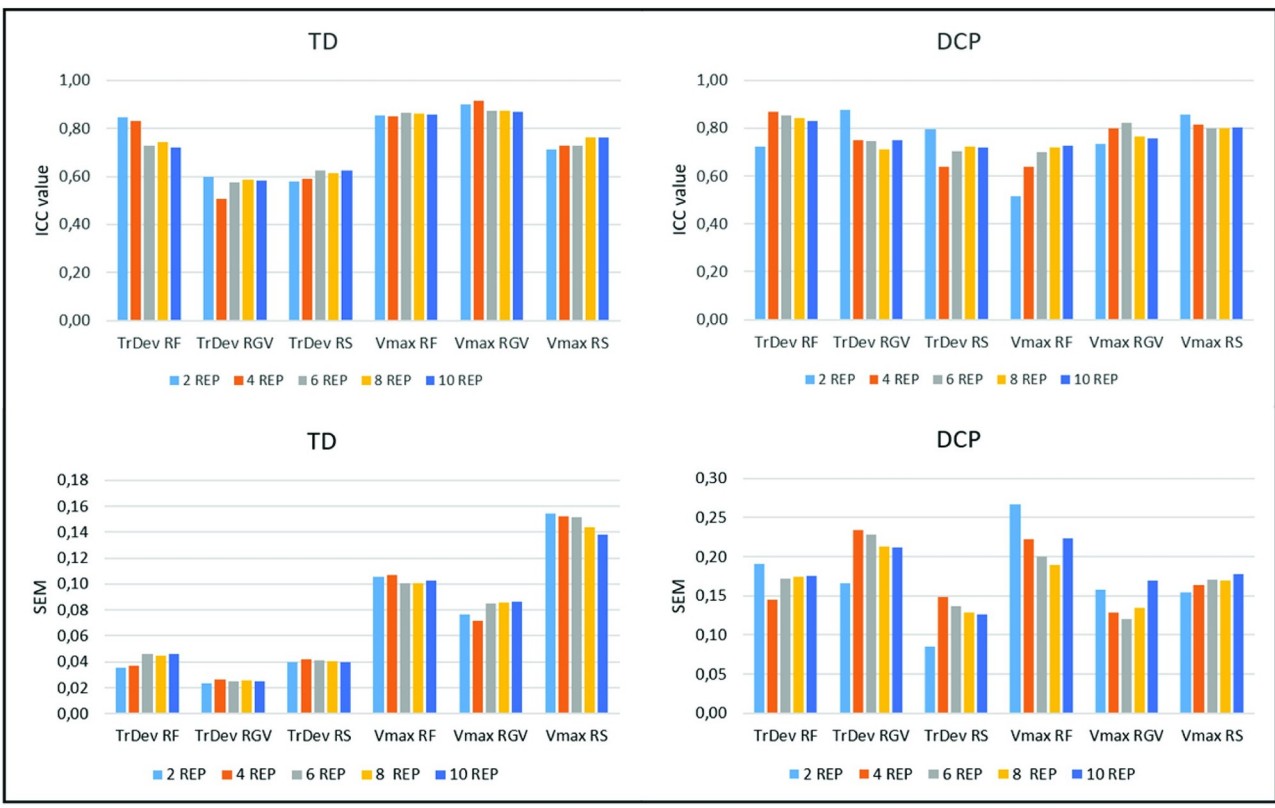

**Fig 4. Within-session Intra-class correlation coefficients (ICC's) and standard error of measurement (SEM) for spatio-temporal parameters for TD participants and participants with DCP.** TrDeV = trajectory deviation; Vmax = maximal velocity; RF = reach forward; RGV = reach and grasp vertical; RS = reach sideways.

### Between-session repeatability

As the SEM values did not change after eight repetitions for the DCP group, variability between sessions will be based on the consistency measure from eight repetitions.

The intrinsic variability between sessions was obtained by comparing the consistency measure of session 1 and session 2 for joint angles at PTA (Fig 6A). Consistency measures were higher for the DCP group in comparison with the TD group for all shoulder angles, elbow pro/supination, wrist flexion/extension, wrist deviation, trunk lateral flexion and trunk axial rotation. Overall, consistency measures were below 10˚ for all angles except wrist flexion/extension for the DCP group, whilst there were no task-dependent differences.

Extrinsic variability was obtained by comparing the mean of the consistency measure of session 1 and 2 with the between-session standard error for joint angles at PTA (Fig 6B). Highest between-session differences were found for elbow pro/supination and scapular angles for both groups. Overall, between-session standard error was below 10˚ except for shoulder rotation, elbow angles, wrist flexion/extension and scapular tilting.

For the spatio-temporal parameters (Fig 7A), intrinsic variability was very similar for session 1 and session 2, with consistency measures below 0.2 for trajectory deviation and maximal velocity for both groups. Both parameters were higher for the DCP group in comparison with the TD participants. For extrinsic variability (Fig 7B), both trajectory deviation and maximal velocity showed higher consistency measures within-session in comparison with between-session, but there were no task-specific differences.

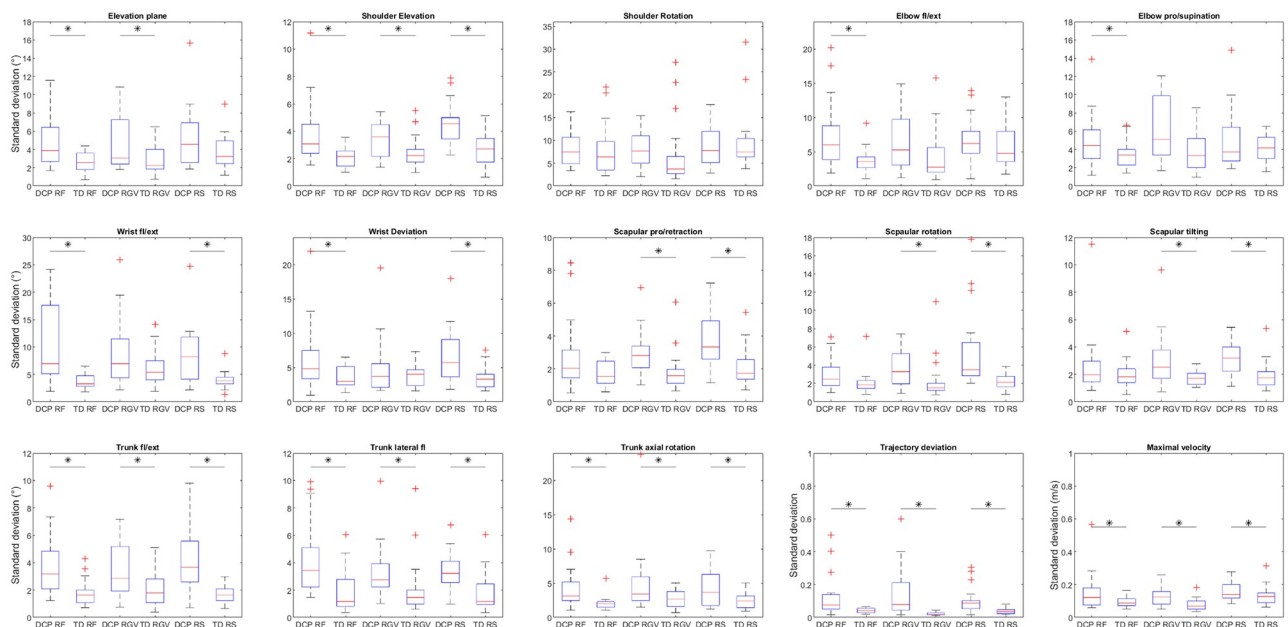

**Fig 5. Boxplots of standard deviations of joint angles at PTA and spatio-temporal parameters for the TD and DCP group.** Fl = flexion;
ext = extension; RF = reach forward; RGV = reach and grasp vertical; RS = reach sideways.

## Between-group differences

Mean and standard deviations of joint angles at PTA and spatio-temporal parameters are
reported in Table 1. During RF, participants with DCP showed higher external shoulder rota-
tion (-46.3˚ versus -31.7˚; p = 0.004), higher elbow flexion (34.4˚ versus 12.9˚; p<0.001) and
higher trunk axial rotation (10.9˚ versus 7.0˚; p = 0.042). During RGV, participants with DCP
showed lower shoulder adduction (73.63˚ versus 89.06˚. p<0.001) and elevation (-71.1˚ versus
-79.6˚; p = 0.006), higher elbow flexion (36.5˚ versus 17.8˚; p = 0.000) and pronation (102.2˚
versus 76.7˚; p = 0.001) and higher trunk axial rotation (15.6˚ versus 9.1˚; p = 0.001). During
RS, participants with DCP showed less shoulder abduction (12.9˚ versus 21.3˚; p = 0.020);
higher external shoulder rotation (-43.8˚ versus -29.32˚; p = 0.008), higher elbow flexion (31.2˚
versus 15.0˚; p = 0.000) and higher elbow pronation (132.0˚ versus 106.2˚; p = 0.000).

The absolute difference between the TD and DCP group exceeded the between-session
standard error for shoulder rotation and elbow flexion/extension during RF (Fig 8). During
RGV, this was the case for elbow flexion/extension, elbow pro/supination and trunk axial rota-
tion and during RS, the absolute difference exceeded the between-session standard error for
elevation plane, shoulder rotation, elbow flexion/extension, elbow pro/supination and wrist
flexion/extension.

For the spatio-temporal parameters, trajectory deviation was significantly higher for partici-
pants with DCP for all tasks (RF: 1.5 versus 1.2; p = 0.002; RGV: 1.4 versus 1.1; p = 0.000; RS:
1.4 versus 1.2; p = 0.000). Similarly, maximal velocity was lower for participants with DCP for
all tasks (RF: 1.0 m/s versus 1.4 m/s; p = 0.000; RGV: 0.9 m/s versus 1.3 m/s; p = 0.000; RS: 1.2
m/s versus 1.6 m/s; p = 0.000).

The absolute difference between the TD and DCP group exceeded the between-session
standard error for all parameters and all tasks (Fig 7C).

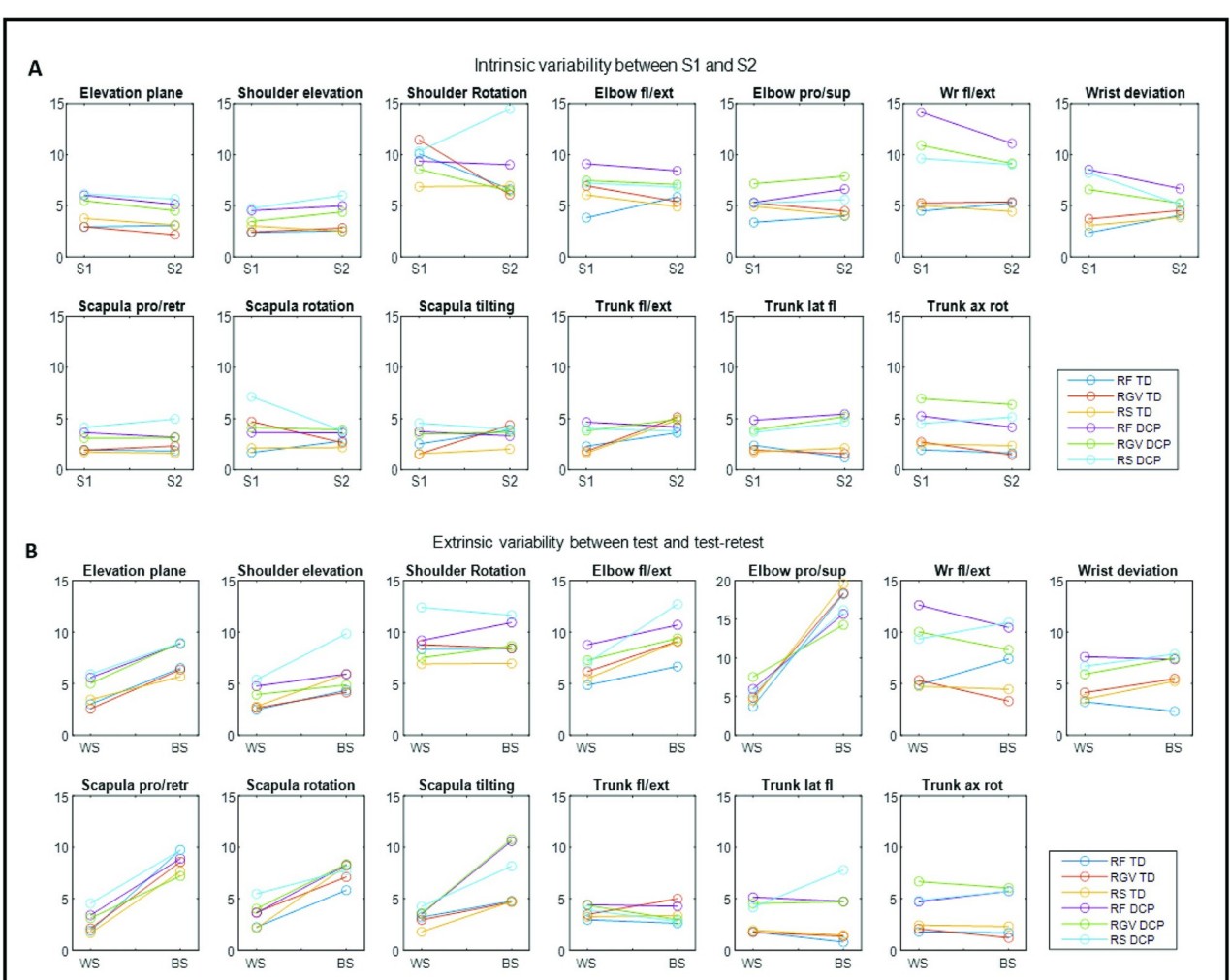

**Fig 6. Intrinsic variability (A) between session 1 (S1) and 2 (S2) and extrinsic variability (B) for within (WS)–and between-session (BS) for participants with DCP and TD participants for all tasks.** Fl = flexion; ext = extension; Wr = wrist; pro = protraction; retr = retraction; lat fl = lateral flexion; ax rot = axial rotation; RF = reach forward; RGV = reach and grasp vertical; RS = reach sideways. (Lines serve for visualisation purposes only).

## Discussion

The aim of this study was to assess the psychometric properties of upper limb kinematics in children and adolescents with and without DCP. Only one study has previously included participants with DCP in a three-dimensional upper limb motion protocol during the reach and grasp cycle [23], but the reliability of the protocol in the abovementioned study has not yet been assessed for participants with DCP. For children with hemiplegia, several studies investigated the reliability of upper limb kinematics, but all of them used a fairly small amount of repetitions, ranging from three [13, 14, 17] to six [18]. In stroke, the number of included repetitions differs between two and 10, with one study evaluating the effect of the number of repetitions on reliability values [31, 32]. As the movement patterns of children and adolescents with DCP are characterized by involuntary movements, it is likely that more repetitions are required to properly capture and describe the most representative patient-specific motion pattern.

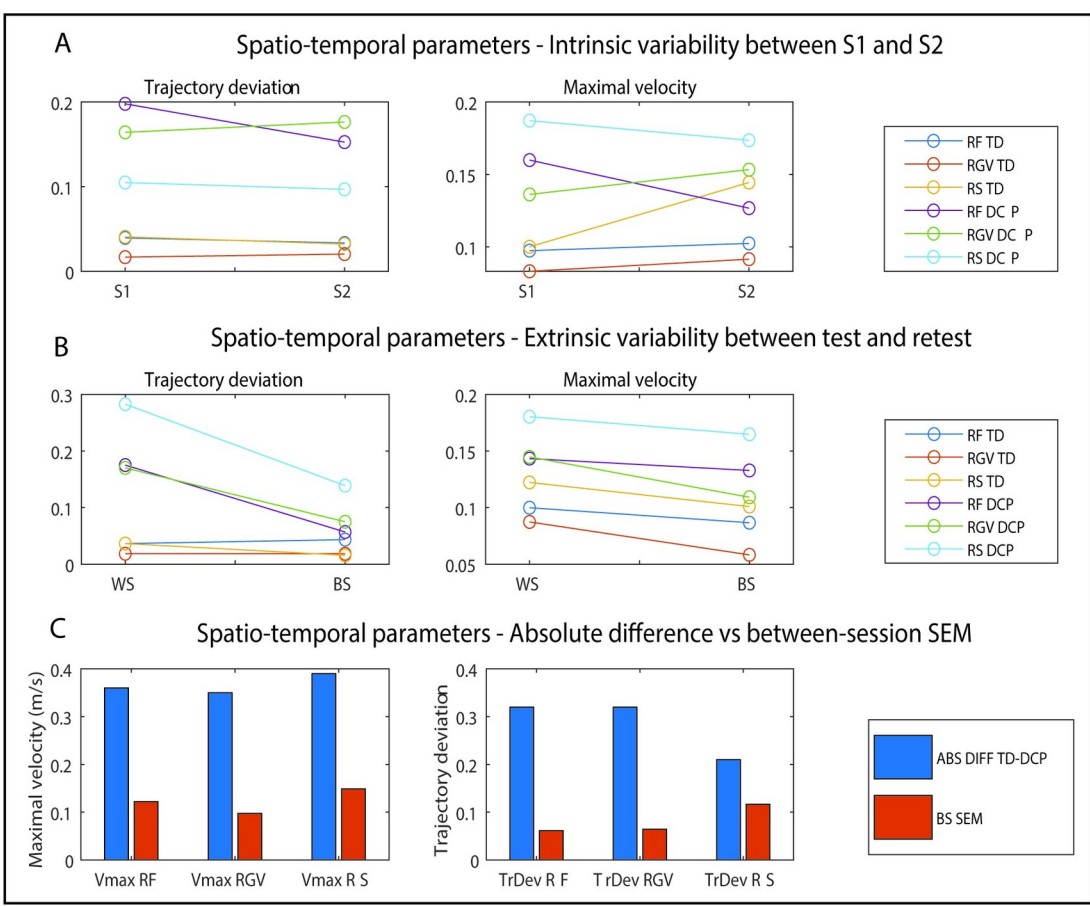

**Fig 7. Intrinsic variability (A), extrinsic variability (B) and the comparison absolute difference and between-session measurement error (C) for spatio-temporal parameters.** RF = reach forward; RGV = reach and grasp vertical; RS = reach sideways. ABS DIFF = absolute difference; BS SEM = between-session measurement error. (Lines serve for visualisation purposes only).

As such, the first goal was to explore how many repetitions of a functional upper limb task should be recorded in order to capture all movement variability when evaluating children and adolescents with DCP. For the joint angles at PTA, ICC values were above 0.60 for all joint angles for the TD participants for all number of repetitions, except for wrist deviation during RF. The lower ICC value for wrist deviation seems to reflect natural variability, as eight out of 20 participants had a range of more than 10° in joint angle at PTA between repetitions in the same session. Furthermore, these findings corroborate previous results in an upper limb kinematics reliability study, showing high within-session reliability for all angles except wrist flexion/extension and deviation [19]. For participants with DCP, ICC values were above 0.60 for all joint angles, except trunk lateral flexion during RF, where the ICC value increased from 0.34 to 0.59 when increasing the number of repetitions from two to six. For the SEM, highest values were found for shoulder rotation in the TD group and wrist flexion/extension in the DCP group, with overall higher SEM values for the DCP group for all joint angles and all tasks. These results are in agreement with Jaspers et al. [18] for children with hemiplegic spastic CP, with slightly higher values for shoulder rotation, elbow flexion/extension and pro/supination and wrist flexion/extension for all tasks in our study. Overall, these results imply that joint angles at PTA are reliable over multiple repetitions within one session, where a minimum of

**Table 1. Mean and standard deviations of joint angles at point of task achievement as well as the p-value for the between-group differences.**

| | REACH FORWARD | | | REACH AND GRASP VERTICAL | | | REACH SIDEWAYS | | |
|---|---|---|---|---|---|---|---|---|---|
| | TD Mean (STD) | DCP Mean (STD) | p-value | TD Mean (STD) | DCP Mean (STD) | p-value | TD Mean (STD) | DCP Mean (STD) | p-value |
| **Elevation plane** | 79.99˚ (6.48) | 73.01˚ (12.72) | 0.021 | 89.06˚ (8.65) | 73.63˚ (11.92) | <0.001** | 12.86˚ (9.40) | 21.29˚ (14.94) | 0.020* |
| **Shoulder elevation** | -75.44˚ (7.47) | -72.27˚ (10.06) | 0.264 | -79.82˚ (7.96) | -71.16˚ (10.39) | 0.006** | -64.38 (8.65) | -63.64 (10.47) | 0.405 |
| **Shoulder rotation** | -31.71˚ (14.33) | -46.34˚ (17.14) | 0.004** | -42.60˚ (15.89) | -52.86˚ (16.87) | 0.421 | -29.32˚ (16.38) | -43.79˚ (19.47) | 0.008** |
| **Elbow flexion/extension** | 12.94˚ (9.20) | 34.43˚ (14.67) | <0.001** | 17.78˚ (8.14) | 36.49˚ (14.48) | <0.001** | 14.97˚ (12.82) | 31.18˚ (16.47) | <0.001** |
| **Elbow pro/supination** | 120.10˚ (21.36) | 124.71˚ (16.75) | 0.228 | 76.67˚ (21.37) | 102.22˚ (23.04) | 0.001** | 106.17˚ (21.29) | 131.98˚ (23.06) | <0.001** |
| **Wrist flexion/extension** | 7.19˚ (10.47) | 16.00˚ (17.40) | 0.030* | -14.67˚ (11.84) | -8.19˚ (21.16) | 0.245 | 6.45˚ (11.99) | 17.69˚ (26.09) | 0.045* |
| **Wrist deviation** | 1.40˚ (4.46) | 1.59˚ (11.44) | 0.473 | -1.02˚ (7.99) | -2.12˚ (13.48) | 0.760 | 3.85˚ (5.28) | 7.59˚ (12.22) | 0.109 |
| **Scapular pro/retraction** | 43.46˚ (9.85) | 39.09˚ (10.48) | 0.091 | 42.87˚ (8.57) | 39.24˚ (8.21) | 0.191 | 17.62˚ (13.70) | 19.43˚ (14.81) | 0.347 |
| **Scapular rotation** | -14.33˚ (12.10) | -13.70˚ (14.38) | 0.441 | -10.60˚ (10.58) | -9.79˚ (15.80) | 0.853 | -15.52˚ (10.46) | -14.32˚ (16.38) | 0.393 |
| **Scapular tilting** | -1.19˚ (10.73) | 1.08˚ (12.39) | 0.270 | 1.04˚ (10.19) | 0.39˚ (13.46) | 0.867 | -1.45˚ (11.65) | 3.65˚ (14.78) | 0.119 |
| **Trunk flexion/extension** | -3.01˚ (4.24) | -0.64˚ (9.40) | 0.155 | -2.34˚ (5.24) | -0.31˚ (8.87) | 0.389 | -1.18˚ (5.86) | -0.82˚ (9.17) | 0.442 |
| **Trunk lateral flexion** | -0.08˚ (4.64) | -2.86˚ (5.98) | 0.055 | -1.22˚ (3.74) | -3.07˚ (5.88) | 0.250 | -0.81˚ (4.36) | 0.13˚ (5.96) | 0.288 |
| **Trunk axial rotation** | 7.00˚ (4.56) | 10.85˚ (8.62) | 0.042* | 9.06˚ (4.67) | 15.58˚ (7.71) | 0.001** | -7.56˚ (4.40) | -13.71˚ (11.03) | 0.013* |
| **Trajectory deviation** | 1.19 (0.08) | 1.51 (0.41) | 0.002** | 1.07 (0.03) | 1.43 (0.40) | <0.001** | 1.15 (0.05) | 1.36 (0.22) | <0.001** |
| **Maximal velocity (m/s)** | 1.36 (0.25) | 1.00 (0.31) | <0.001** | 1.25 (0.23) | 0.90 (0.25) | <0.001** | 1.63 (0.26) | 1.24 (0.34) | <0.001** |

TD = typically developing; DCP = dyskinetic cerebral palsy;

* = p-value < 0.05;

** = p-value < 0.01

eight repetitions is advised for both TD participants and participants with DCP. This is higher than a similar approach in stroke, where the main result was that 3 repetitions was sufficient for the majority of the kinematic parameters during a drinking task [32]. The lack of task-specific differences in both ICC and SEM values implies that the variability we found is truly intrinsic due to the fluctuating movements in dyskinetic CP, rather than task-specific or methodological.

For the spatio-temporal parameters, maximal velocity was shown to be a reliable parameter for both groups for all tasks, whereas trajectory deviation showed good reliability for participants with DCP but somewhat lower values for TD participants during RGV and RS. Overall, the variability in trajectory deviation between participants with DCP (i.e., values ranging between 1.1 and 3.3) was much higher compared to TD participants (1.0–1.5) for all tasks. It is thus possible that the lower ICC values in the TD participants are the consequence of a small range in trajectory deviation values between participants, as the ICC value is known to perform less well in the absence of variability [30]. SEM values were higher for the DCP group in comparison with the TD group, and higher for maximal velocity in comparison with trajectory deviation for both groups.

The second goal was to compare variability between the TD and DCP group using standard deviations. For the majority of the angles and tasks, the participants with DCP showed more variability in comparison with their TD peers. This finding agrees with the hypothesis that individuals with DCP present with a more variable movement pattern with less consistency over multiple repetitions. In the wrist joint, this variability was task-dependent as the variability between both groups was significantly different during RF and RS, but not during RGV. This may be a consequence of the more specific grasping movement during RGV, allowing for somewhat less variability during task execution while additionally requiring wrist extension. Similar to the wrist joint, the DCP group showed higher variability at the level of the scapula

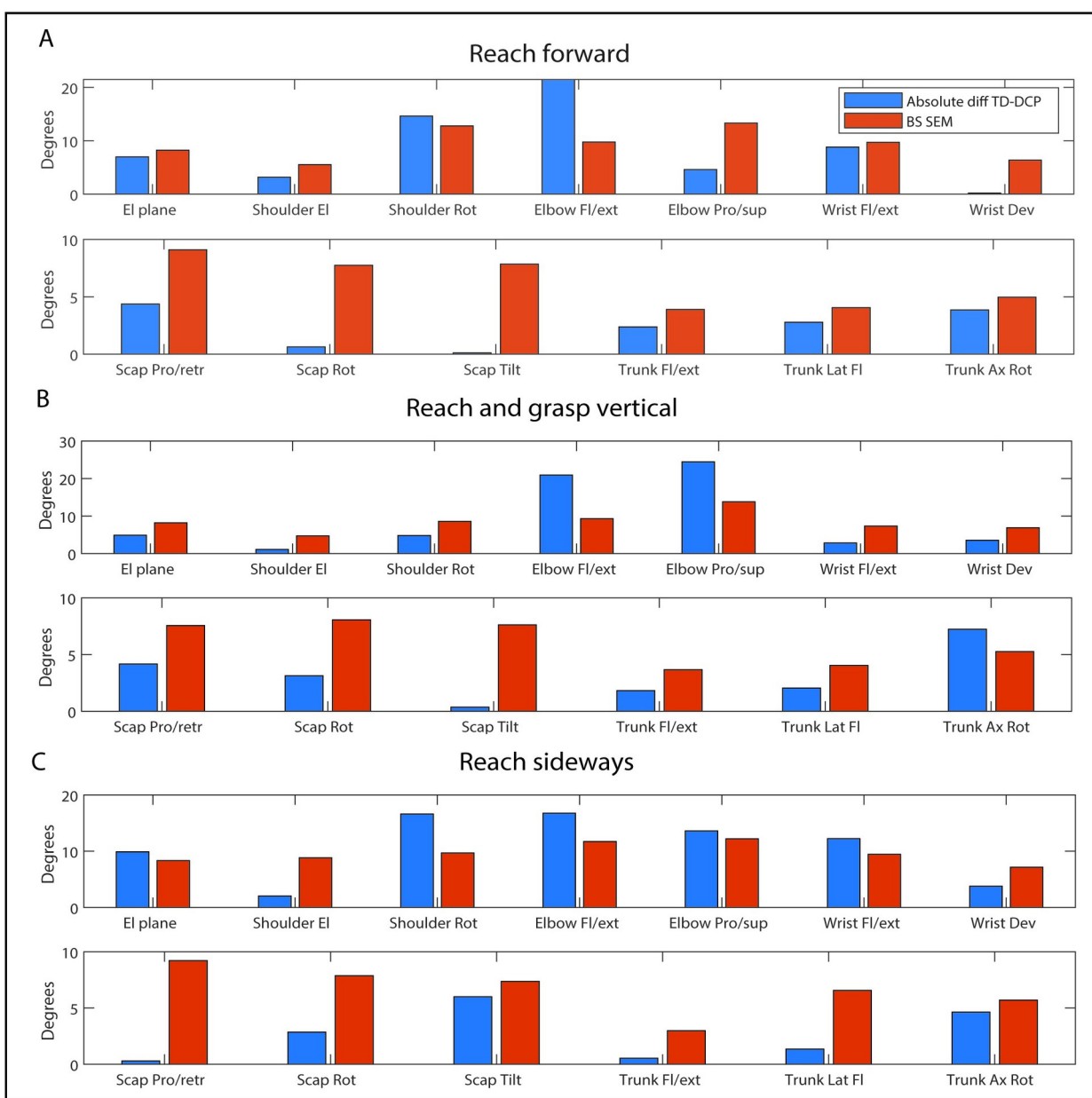

**Fig 8. Absolute difference (TD vs DCP) and between-session measurement error for joint angle at PTA.** El = elevation; Rot = rotation; Fl = flexion; Ext = extension; Dev = deviation; Pro = protraction; Scap = scapula; Retr = retraction; Rot = rotation; Tilt = tilting; Lat fl = lateral flexion. Absolute diff = absolute difference. BS SEM = between-session measurement error.

during RGV and RS, but not RF. During RS, higher variability could be the consequence of movement outside of the sagittal plane while during RGV, scapula movement may be the consequence of trunk compensation for incomplete wrist and elbow extension.

The third goal was to assess between-session repeatability of these joint angles and spatio-temporal parameters, as this is an important first step towards the definition of responsiveness of these measures. Intrinsic variability between-sessions was evaluated by comparing the consistency measure (based on eight repetitions) from session 1 and session 2. Extrinsic variability

was evaluated by comparing the mean consistency measure of session 1 and 2 with the between-session SEM value.

Intrinsic variability showed few differences between sessions for both the TD and DCP group for both joint angles and spatio-temporal parameters, implying that intrinsic variability is rather low, even in participants with DCP. When comparing consistency over time, factors such as fatigue and a learning effect could possibly influence time-dependent performance. With a learning effect, one would expect a more consistent performance during the second session, while fatigue is expected to decrease the consistency over time. It is thus possible that both effects are present, but neither has a significant effect because of their counterbalancing impact or that both effects are significant, but the overall effect is not.

Extrinsic variability showed larger differences between test and test-retest, indicating that methodological aspects influence the results to a certain extent. Specifically for elbow pro/supination and scapula angles, SEM values were higher between-sessions in comparison with within-session SEM values. These differences imply that the influence of marker placement, palpation differences and positioning should be taken into account when interpreting the results of upper limb kinematics between different sessions. The higher between-session SEM values and lower ICC values for the scapula joint agree with previous findings in hemiplegic spastic CP and healthy adults, all presenting lower reliability and higher errors between-sessions which implies that caution is warranted when interpreting these joint angles over a longer period of time [18, 33]. A recent review confirmed lower test-retest reliability in multiple studies, indicating that when an accromion cluster is used, multiple calibrations can improve measurement accuracy [34]. For the spatio-temporal parameters, both intrinsic and extrinsic variability were low, implying that both are reliable parameters to assess within and between sessions. The good between-session results for the majority of the parameters allow for a reliable measurement of upper limb movement characteristics before and after intervention, or for longitudinal follow-up of patients.

The fourth goal of this study was to assess the differences in upper limb kinematics between children and adolescents with and without DCP, reflecting the construct validity of the test protocol. For the joint angles at PTA, elbow flexion/extension and pro/supination were most discriminative between participants with and without DCP, as participants with DCP showed higher elbow flexion and pronation in all three tasks, except for elbow pronation during RF. These results are in agreement with previous findings for children with hemiplegic spastic CP [20–22]. Additionally, wrist flexion was significantly higher in participants with DCP during RF and RS, which is in agreement with Jaspers et al. [20]. However, wrist flexion was not significantly different between both groups during RGV, which is in contradiction with the previous results [20]. This finding agrees with the clinical presentation that children and adolescents with DCP have less range of motion restrictions but rather have a more variable kinematic pattern due to intermittent muscle contractions [5]. Trunk axial rotation was higher for participants with DCP in comparison with the TD participants for all tasks. Trunk compensation in CP during grasping and sideways reaching has been reported before [13, 20, 35] and is described as a compensation mechanism for kinematics deficits at the level of the elbow and wrist. However, since none of our participants showed passive range of motion deficits and DCP is characterized by variable movement patterns, it is possible that our participants used these compensation strategies, but less frequently than their peers with spastic CP.

For a selection of joint angles, the absolute difference between TD and DCP exceeded the between-session standard error, which implies that those angles are most sensitive when groups. This is a clinically important finding as for those specific joint angles, the clinical pattern between TD participants and participants with DCP is more distinct than the measurement error occurring over time. This pattern was task-specific, with the most sensitive joint

angles during RF being shoulder rotation and elbow flexion/extension. During RGV, elbow flexion/extension and pro/supination as well trunk axial rotation were most sensitive. During RS, elevation plane, shoulder rotation, elbow flexion/extension and pro/supination as well as wrist flexion were most sensitive. For wrist flexion during RF, the absolute difference TD-DCP was only slightly smaller than the between-session standard error, and since wrist flexion was significantly different between the TD and DCP group, this angle should also be considered a parameter of interest during RF. For the spatio-temporal parameters, the difference between the TD and DCP group was much higher than the between-session measurement error for both maximal velocity and trajectory deviation, implying that both are reliable and discriminative parameters between TD participants and participants with DCP, which is in line with previous results in hemiplegic spastic CP [16, 35] and DCP [12]. Overall, these differences show that the abnormal postures caused by intermittent muscle contractions interfere with typical movement and that joint angles at PTA and trajectory deviation and maximal velocity are useful parameters to assess these differences.

This study warrants some critical reflections. As DCP is much less prevalent than spastic CP and the children with DCP who have sufficient reaching and grasping possibilities to execute functional upper limb tasks are a minority within the DCP group, the age range of the participants was large, as it is not straightforward to obtain a large sample size. Multicentre efforts may help to increase the sample size in future studies, allowing for a larger representation of dyskinetic movement patterns and an age-related group analyses. Due to the large age range, a distinction could not be made between immature and mature movement patterns, as the latter were previously found to occur at the age of 12 years old [36]. Second, the high ratio between-within session errors for the scapula joint means that palpation inaccuracies at the level of the scapula influence the scapulothoracic joint angles to a large extent, suggesting that these angles may be slightly less reliable when evaluated over time. A possibility to improve errors in future measurements could be to palpate the anatomical landmarks of the scapula twice and use an averaged value of two palpations for each session.

## Conclusion

This is the first study to report the psychometric properties of upper limb kinematics in participants with DCP during functional tasks, yielding excellent within-session, moderate to good between-session repeatability and good content validity. Thereby, this study provides a first step towards an optimisation of the current treatment strategies by examining the movement patterns using objective features in children and adolescents with DCP. As previous studies mainly focused on upper limb kinematics in children with spastic hemiplegic CP, our results emphasized the importance of a personalized approach for children and adolescents with DCP. We advise to include a minimum of eight repetitions when recording the upper limb movement patterns in children and adolescents with DCP, as to capture the full variability present in the movement pattern when executing functional tasks.

## Supporting information

**S1 Table. Participant characteristics.** TD = typically developing; DCP = dyskinetic cerebral palsy; MACS = manual ability classification system; M = male; F = female.
(DOCX)

**S2 Table. Intra-class correlation coefficients (ICC) and standard error of measurement (SEM) and change in ICC and SEM values with each number of different repetitions, expressed in %.** TD = typically developing participants; DCP = participants with dyskinetic

cerebral palsy. Fl = flexion; Ext = extension; Pro = protraction; Retr = retraction; Rot = rotation, REP = repetitions.
(DOCX)

**S3 Table. Intra-class correlation coefficients (ICC) and standard error of measurement (SEM) and change in ICC and SEM values with each number of different repetitions, expressed in %.** TD = typically developing; DCP = dyskinetic cerebral palsy; Vmax = maximal velocity, REP = repetitions.
(DOCX)

**S4 Table. Median and interquartile ranges of standard deviations for joint angles at point of task achievement and spatio-temporal parameters as well as the p-value for the between-group differences.** TD = typically developing; DCP = dyskinetic cerebral palsy; IQR = inter-quartile range $^*$ = p-value $<$ 0.05; $^{**}$ = p-value $<$ 0.01.
(DOCX)

**S1 Data.**
(XLSX)

## Acknowledgments

The author would like to thank the participants and their parents for the time and effort to participate in this study.

## Author Contributions

**Conceptualization:** Inti Vanmechelen, Hilde Feys, Kaat Desloovere, Jean-Marie Aerts, Elegast Monbaliu.

**Data curation:** Inti Vanmechelen, Elegast Monbaliu.

**Formal analysis:** Inti Vanmechelen, Kaat Desloovere.

**Funding acquisition:** Elegast Monbaliu.

**Investigation:** Inti Vanmechelen.

**Methodology:** Inti Vanmechelen, Saranda Bekteshi, Marco Konings, Kaat Desloovere, Elegast Monbaliu.

**Project administration:** Elegast Monbaliu.

**Resources:** Kaat Desloovere, Elegast Monbaliu.

**Software:** Inti Vanmechelen.

**Supervision:** Elegast Monbaliu.

**Visualization:** Inti Vanmechelen.

**Writing – original draft:** Inti Vanmechelen.

**Writing – review & editing:** Inti Vanmechelen, Saranda Bekteshi, Marco Konings, Hilde Feys, Kaat Desloovere, Jean-Marie Aerts, Elegast Monbaliu.

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
