## [Decision Letter · Decision Letter 0]

19 Jul 2022

PONE-D-22-07917Psychometric properties of upper limb kinematics during functional tasks in children and adolescents with dyskinetic cerebral palsyPLOS ONE

Dear Dr. Vanmechelen,

Thank you for submitting your manuscript to PLOS ONE. After careful consideration, we feel that it has merit but does not fully meet PLOS ONE’s publication criteria as it currently stands. Therefore, we invite you to submit a revised version of the manuscript that addresses the points raised during the review process.

We look forward to receiving your revised manuscript.

Kind regards,

Yih-Kuen Jan, PhD

Academic Editor

PLOS ONE

“IV received an FWO fellowship. The funder had no involvement in study design, collection, analysis and interpretation of data, writing of the report, or in the decision to submit the article for publication.”

“IV is funded by Fonds Wetenschappelijk onderzoek Vlaanderen (FWO), grant number 65831. https://www.fwo.be/. The funder did not play a role in the study design, data collection and analysis, decision to publish, or preparation of the manuscript.”

Reviewers' comments:

Reviewer's Responses to Questions

**Comments to the Author**

1. Is the manuscript technically sound, and do the data support the conclusions?

Reviewer #1: Yes

Reviewer #2: Partly

2. Has the statistical analysis been performed appropriately and rigorously? 

Reviewer #1: Yes

Reviewer #2: Yes

3. Have the authors made all data underlying the findings in their manuscript fully available?

Reviewer #1: Yes

Reviewer #2: Yes

4. Is the manuscript presented in an intelligible fashion and written in standard English?

Reviewer #1: Yes

Reviewer #2: No

5. Review Comments to the Author

Reviewer #1: GENERAL

The authors have performed an observational study in which they evaluated clinimetric properties of upper limb kinematics during three functional tasks with an optimal motion capture system in 20 children and adolescents with dyskinetic cerebral palsy and 20 typically developing children. They found varying ICC values and kinematic parameters showed the variability in children with DCP.

The manuscript is well structured, and easy to read. There are however a couple of aspects that need revision:

MAJOR COMMENTS

1. When performing research in the clinimetric properties domain, the COSMIN standards should be mentioned and these should be applied (https://www.cosmin.nl/). Please adapt the manuscript accordingly.

2. In the methods section, the model used to determine the ICCs should be described. In addition, the cut-off for an acceptable ICC should be mentioned, including relevant references.

3. The discussion is long, but there is relatively little referencing to other work. It is hard to find references within DCP, but the question is, whether a comparison couldn’t be made with other neurological diseases? For example:

• What is recommended in stroke regarding the number of repetitions in kinematic assessments?

• What is known in the literature regarding between-session errors when measuring scapula movements?

OTHER COMMENT

1. Please explicitly mention the number of recommended repetitions in the conclusion of the abstract, as this is an important message based on the study findings.

Reviewer #2: - The major issue in this study is the small sample size "20 children" that couldn't provide significant findings.

Comments:

Abstract:

1. Trial design is not mentioned.

2. Methods section is poorly framed. It has to be re-written.

Introduction:

1. Explain the rationale of the study. Kindly focus on three elements of introduction.

a. What is known about the topic? (Background)

b. What is not known? (The research problem)

c. Why the study was done? (Justification)

2. Objectives and hypothesis are not clear.

Methods:

1. Methods section determines the results. Kindly focus on three basic elements of methods section.

a. How the study was designed?

b. How the study was carried out?

c. How the data were analyzed?

d. Components (SPICES) for methods

i. Study design, setting, sample size

ii. Participant

iii. Intervention/issue of interest (exposure)

iv. Comparison

v. Ethics and end point

vi. Statistical analysis

2. Also please do take a reference from the checklist so that the paper is scientifically sound.

a. Describe the trial design and allocation ratio.

b. What changes were made to methods after trial commencement? Kindly explain with reasons.

c. What were Eligibility criteria for participants?

d. Mention the settings and locations where the data were collected.

e. Define pre-specified primary and secondary outcome measure.

f. How was sample size determined?

g. Who enrolled participants?

h. Who assigned participants to interventions?

i. Statistical methods need more explanation.

6. PLOS authors have the option to publish the peer review history of their article (what does this mean?). If published, this will include your full peer review and any attached files.

Reviewer #1: No

Reviewer #2: **Yes: **Walid Kamal Abdelbasset

---

## [Author Response · Author response to Decision Letter 0]

25 Aug 2022

Response to reviewers

Note from the author: Throughout the rebuttal letter, the replies to the reviewers/editors comments will be given in italic. The adapted/added text in the manuscript will additionally be given, including page and line numbers.

Editor’s comments to the author

Reply: File naming and manuscript style have been adjusted to apply to the PLOS ONE requirement according to the guidelines above.

“IV received an FWO fellowship. The funder had no involvement in study design, collection, analysis and interpretation of data, writing of the report, or in the decision to submit the article for publication.”

“IV is funded by Fonds Wetenschappelijk onderzoek Vlaanderen (FWO), grant number 65831. https://www.fwo.be/. The funder did not play a role in the study design, data collection and analysis, decision to publish, or preparation of the manuscript.”

a

Reply: We have removed the funding section from the manuscript. The funding section in the online submission form is correct and does not need to be amended. This is equally mentioned in the cover letter.

Reviewers' comments:

Reviewer's Responses to Questions

Comments to the Author

1. Is the manuscript technically sound, and do the data support the conclusions?

Reviewer #1: Yes Reviewer #2: Partly

2. Has the statistical analysis been performed appropriately and rigorously? 

Reviewer #1: Yes Reviewer #2: Yes

3. Have the authors made all data underlying the findings in their manuscript fully available?

Reviewer #1: Yes Reviewer #2: Yes

4. Is the manuscript presented in an intelligible fashion and written in standard English?

Reviewer #1: Yes Reviewer #2: No

Reply: We have thoroughly revised the manuscript and corrected typographical/grammatical errors.

Review Comments to the Author

Reply: Thank you for the opportunity to adapt our manuscript towards the reviewers comments. Please find below the response to the reviewers’ comments and the adapted text in the manuscript.

Reviewer #1: GENERAL

The authors have performed an observational study in which they evaluated clinimetric properties of upper limb kinematics during three functional tasks with an optimal motion capture system in 20 children and adolescents with dyskinetic cerebral palsy and 20 typically developing children. They found varying ICC values and kinematic parameters showed the variability in children with DCP.

The manuscript is well structured, and easy to read. There are however a couple of aspects that need revision:

MAJOR COMMENTS

1. When performing research in the clinimetric properties domain, the COSMIN standards should be mentioned and these should be applied (https://www.cosmin.nl/). Please adapt the manuscript accordingly.

Reply: Thank you for this suggestion. The Cosmin checklist was reviewed and used further throughout the manuscript. This has been added on P.5, line 139-40:

“The Cosmin checklist was used for standardisation of reporting of clinimetric properties (1).”

The Cosmin items that were not in the manuscript yet have been added:

P.5 line 146-153: Study design has been added. “Study design Within-session reliability and repeatability were evaluated using the intra-class correlation coefficient and standard error of measurement on the parameters collected within one session. Between-session repeatability was evaluated by using data of the first and second session, and intrinsic and extrinsic variability were explored. All parameters were compared between the TD individuals and individuals with DCP to evaluate between-group differences.”

P.6 line 156-158: “Individuals with DCP were eligible to participate if they: were diagnosed with DCP by a paediatric neurologist” has been added to the eligibility criteria.

P.9 line 219-220: Handling of missing data has been added: “In case of missing data, the joint angle for which the data is missing was excluded from the subsequent data analysis for this specific participant.”

P.10/11 line 267-273: Different sample sizes according to missing values have been added to the results section: aaaaaaaaaaaaaaaaaa “Two participants from the DCP group were unable to perform the reach and grasp vertical task and one participant with DCP did not perform the reach sideways task due to fatigue. For four participants (2 TD; 2 DCP), the values for shoulder rotation and elevation plane during the reach forward and reach and grasp vertical task were incorrect and removed from the analyses. The ICC and SEM values for reach and grasp vertical are thus based on 18 participants with DCP for all angles except for elevation plane and shoulder rotation (16 participants) and 18 TD participants. The ICC and SEM values for reach sideways are based on 19 participants with DCP and 20 TD participants.”

a

2. In the methods section, the model used to determine the ICCs should be described. In addition, the cut-off for an acceptable ICC should be mentioned, including relevant references.

Reply: Thank for raising this point. The model used to determine the ICCs and the cut-off values was added on p.9 line 222-228:

“Subsequently, intra-class correlation coefficient (ICC) values – ICCw(2,1) based on single measures – and standard error of measurement (SEM) were calculated. … Values of ICC were interpreted as poor (<0.50), moderate (0.50- < 0.75), good (0.75–0.90), and excellent (> 0.90) (2).”

a

3. The discussion is long, but there is relatively little referencing to other work. It is hard to find references within DCP, but the question is, whether a comparison couldn’t be made with other neurological diseases? For example:

• What is recommended in stroke regarding the number of repetitions in kinematic assessments?

• What is known in the literature regarding between-session errors when measuring scapula movements?

Reply: Thank you for this helpful comment. We have indeed been able to make a comparison with stroke, which is added in the discussion on P.15, line 352-354:

“For children with hemiplegia, several studies investigated the reliability of upper limb kinematics, but all of them used a fairly small amount of repetitions, ranging from three (3-5) to six (6). In stroke, the number of included repetitions differs between two and 10, with one study evaluating the effect of the number of repetitions on reliability values (7, 8).”

And additionally on P.16, line 374-377:

“Overall, these results imply that joint angles at PTA are reliable over multiple repetitions within one session, where a minimum of eight repetitions is advised for both TD participants and participants with DCP. This is higher than a similar approach in stroke, where the main result was that 3 repetitions was sufficient for the majority of the kinematic parameters during a drinking task (8).”

We have additionally added some more references on between-session reliability and measurement error for the scapula joint on P.18, lines 417-422:

“The higher between-session SEM values and lower ICC values for the scapula joint agree with previous findings in hemiplegic spastic CP and healthy adults, all presenting lower reliability and higher errors between-sessions which implies that caution is warranted when interpreting these joint angles over a longer period of time (6, 9). A recent review confirmed lower test-retest reliability in multiple studies, indicating that when an accromion cluster is used, multiple calibrations can improve measurement accuracy (10).”

OTHER COMMENT

1. Please explicitly mention the number of recommended repetitions in the conclusion of the abstract, as this is an important message based on the study findings.

Reply: The number of recommended repetitions has been added in the conclusion section of the abstract:

“This is the first study to assess the psychometric properties of upper limb kinematics in children and adolescents with DCP, showing that children with DCP show higher variability during task execution, requiring a minimum of eight repetitions.”

Reviewer #2: 

The major issue in this study is the small sample size "20 children" that couldn't provide significant findings.

Reply: We understand the reviewer’s concern on the small sample size. However, it should be noted that dyskinetic cerebral palsy (CP) is a rather rare disease. The prevalence of children born with CP in Western Europe is 1.5/1000 (11). Of this group, only 15% is diagnosed with dyskinetic CP, reducing its prevalence to 2.25 per 10 000 live births (12). The number of individuals diagnosed with spastic CP is five times higher in comparison with dyskinetic CP, but our study sample size is bigger than (4, 6, 13) or similar to (14) previous studies in spastic cerebral palsy (CP). This is the first study specifically focusing on dyskinetic CP, and more specifically on upper limb movement patterns in dyskinetic CP, which additionally reduces our goal population, since 50% of the individuals with dyskinetic CP does not have sufficient arm-hand function to be included in this study. Despite this sample size, we were able to identify significant between-group differences, which is highlighted on p.13 lines 324-332 and p.15 lines 334-345.

Comments:

Abstract:

1. Trial design is not mentioned.

2. Methods section is poorly framed. It has to be re-written.

Reply: Thank you for these comments. Trial design has been added to the methods section and the methods section has been critically inspected and discussed with the co-authors. We have re-written the section to improve readability. The methods section now reads:

“In current repeatability and validity study, forty individuals with typical development (n=20) and DCP (n=20) performed a reach forward/sideways and a reach and grasp task during motion analysis on two occasions. Joint angles at point of task achievement (PTA) and spatio-temporal parameters were evaluated within-and between sessions using intra-class correlation coefficients (ICC) and standard error of measurement (SEM). Independent t-tests/Mann-Whitney-U tests were used to compare parameters between groups.” 

Introduction:

1. Explain the rationale of the study. Kindly focus on three elements of introduction.

a. What is known about the topic? (Background)

b. What is not known? (The research problem)

c. Why the study was done? (Justification)

Reply: Thank you for aiming to obtain more structure in the introduction. What is known about the topic is explained on P.4 from lines 97 to 109. We specifically focus on the objective measures in CP:

“Over the past years, there have been several attempts to establish objective measurements in the CP population. Gordon et al., (15) attempted to discriminate dystonia and spasticity in the arm where spasticity was expressed as the amount of force necessary to passively extend the elbow joint as measured with a rigidity analyser and dystonia was characterized as the amount of overflow movement in the contralateral arm. However, evaluating the amount of dystonia only by overflow movements of the contralateral arm does not capture the full aspect of dystonia and its action-specific aspect. Sanger et al., (16) demonstrated an increased movement variability and a lack of straight-line trajectories in participants with DCP during outward reaching. While these results indicate the ability to quantitatively measure movement characteristics of the upper limb using position diodes attached to eight points of the body, they do not provide any information regarding joint angles or movement patterns. When focusing on hemiplegic spastic CP, several upper limb protocols have been developed and validated over the past years (3-5, 17, 18). While all studies presented moderate to good results, the upper limb joints included in the analyses were limited to trunk, shoulder, elbow and wrist angles. The study of Jaspers et al. was the only protocol so far that has additionally presented scapular angles, allowing to investigate the role of the scapula position in upper arm movements (6, 19).”

What is not known is explained on P.4 from line 114 to 118, where we have added more information to emphasize this part of the introduction:

“To date, only one study on kinematic analysis of upper limb movements included children with DCP, representing only a small sub-group of the patient cohort (13). We currently do not know anything about the movement patterns in individuals with DCP as recorded with three-dimensional motion analysis, which currently prohibits us in using this methodology to evaluate the effect of rehabilitation strategies.”

The justification of this study focuses primarily on the increase in insights into the involuntary movement of individuals with DCP, and the fact that evaluation of reliability in crucial before implementing this methodology in routine clinical practice. This is elaborated on P.4 lines 120-123 and P.5 lines 125-127:

“As dyskinetic CP is characterized by involuntary movements, it is expected that their movement patterns will be less consistent compared to TD children or children with spastic CP. In this perspective, we strive towards reliably capturing a pattern that is inherently inconsistent, which may thus require a higher number of repetitions within one session before parameter calculation. Since novel assessments need to be reliable and valid before they can be transferred to clinical practice, the objective of this study is to evaluate the psychometric properties of upper limb kinematics in children and adolescents with and without DCP.”

2. Objectives and hypothesis are not clear.

Reply: This study focuses on one general objective: Evaluation of the psychometric properties of upper limb kinematics in children and adolescents with and without DCP. The subsequent sub-goals and associated hypotheses focus on repeatability, variability and discriminative validity. The text in the manuscript has been rephrased to improve interpretation, P.5 lines 127-143:

“The first goal focuses on repeatability, where the objective is to define the within-session repeatability of joint angles and spatio-temporal parameters and to explore the number of repetitions that are necessary within one session to obtain a representative and robust representation of the movement pattern for participants with and without DCP. The hypothesis is that a higher number of repetitions in comparison with spastic CP is necessary for a robust representation (16). The second goal focuses on the increased variability in the movement patterns of individuals with DCP. The objective is to assess the variability between TD participants and participants with DCP for this specified number of repetitions. We hypothesize that participants with DCP show higher variability in comparison with their TD peers. The third goal focuses on between-session measures. The objective is to assess between-session repeatability of the joint angles and spatio-temporal parameters, as this is an important first step toward responsiveness of these measures. The hypothesis is that joint angles and spatio-temporal parameters can be reliably captured over time. The fourth goal focuses on validity. The objective is to evaluate discriminative validity of three-dimensional motion measures, defining the differences in upper limb kinematics between children and adolescents with and without DCP. The hypothesis is that the joint angles and spatio-temporal parameters will differ significantly between the TD and DCP group.”

Methods:

1. Methods section determines the results. Kindly focus on three basic elements of methods section.

a. How the study was designed?

b. How the study was carried out?

c. How the data were analyzed?

d. Components (SPICES) for methods

i. Study design, setting, sample size

ii. Participant

iii. Intervention/issue of interest (exposure)

iv. Comparison

v. Ethics and end point

vi. Statistical analysis

Reply: Thank you for pointing out the optimal structure for the methodology. We have adapted the methodology section towards this feedback and we have implemented the SPICES method.

a. We have added a section ‘Study design’: P.5 lines 146-153:

“Within-session reliability and repeatability were evaluated using the intra-class correlation coefficient and standard error of measurement on the parameters collected within one session. Between-session repeatability was evaluated by using data of the first and second session, and intrinsic and extrinsic variability were explored. All parameters were compared between the TD individuals and individuals with DCP to evaluate between-group differences. The Cosmin checklist was used for standardisation of reporting of clinimetric properties and we adhered to the SPICES method to ascertain inclusion of all aspects of the methodology (1, 20).”

b. To address the question “how was the study carried out?” and to improve readability, the headings ‘movement protocol’, ‘kinematic model’ and ‘test procedure’ have been merged in the section ‘study procedures’, P.6/7 lines 166-204: 

“Every child was evaluated twice on the same day with a minimum of one hour and a maximum of two hours between sessions at the WE-lab for Health, Technology and Management (KU Leuven, campus Bruges) or the Clinical Movement Analysis Laboratory (CMAL, UZ Leuven, Pellenberg) by the same assessors. All participants were asked to perform three upper limb tasks: reaching forward (RF), reaching sideways (RS) and reach and grasp vertical (RGV). RF, RS and RGV were executed at shoulder height (acromion) and reaching distance was determined according to arm length (from acromion to caput metacarpal III). All tasks were performed at self-selected speed with the non-preferred arm (the hemiplegic arm in participants with unilateral DCP and the non-preferred arm in TD participants) and with both arms in participants with bilateral DCP. Start position (the ipsilateral knee) was indicated with an elastic band above the knee. Every task was executed 10 times per trial with a total of three trials for every task. Participants were seated in a chair with adjustable height and a custom-made reaching system was developed to perform the tasks in a standardized way (Fig 1). The reference position was 90° flexion in hip and knees and the hands placed on the ipsilateral knee (19). Seventeen reflective markers were placed over the body in 5 clusters: two cuffs of 4 markers were placed respectively on the upper arm and forearm, one cluster of 3 markers was placed on the hand and two tripods with 3 markers were placed respectively on the trunk and the scapula. Five segments were thus included (trunk, scapula, humerus, forearm, hand) and four joints were considered (scapulothoracic (scapula), humerothoracic (shoulder), elbow, wrist). Anatomical landmarks were palpated according to precise definitions and digitized using a pointer with four linear markers (19) and anatomical coordinate systems and joint rotation sequences were defined according to the ISB-guidelines (21). Static and dynamic calibrations were subsequently performed for the calculation of anatomical landmarks, during which passive assistance was given where needed. 3D marker tracking was done with 12 infra-red Vicon optical motion capture cameras sampling at 100 Hz and 2 high-definition video cameras, with a typical measurement error of 0.4 mm (Vicon Motion Systems, Oxford Metrics, UK). The currently used protocol has been previously validated in TD participants and participants with hemiplegic spastic CP (6, 19).”

c. How the data were analysed is discussed under the section ‘data analysis’ on P.8, lines 206-220:

“Movement cycles were identified and segmented in Vicon Motion Capture System. One movement cycle was defined from hand on ipsilateral knee to point of task achievement (PTA), where PTA is considered the final point of the reaching or reach-and-grasp cycle (19). The first and last movement cycles were disregarded as they could be influenced by stop and start strategies, resulting in 8 repetitions for each trial, with a total of 24 repetitions for each task. Joint angle at PTA was obtained by selecting the last value of the angular waveform for the joint angles. Subsequently, maximal velocity and trajectory deviation were obtained for each repetition. Trajectory deviation is a dimensionless parameter, but a value of 1 implies a perfect straight line trajectory, whereas the higher the trajectory deviation, the more the movement deviates from a straight line.

To evaluate how many repetitions of a task execution represented a stable movement pattern, an incremental number of repetitions was randomly selected for each task and the change in outcome values was evaluated for both TD participants and participants with DCP.

If missing data will occur, the joint angle for which the data is missing will be excluded from the subsequent data analysis for this specific participant.”

d. The methods section has been adapted to reflect the SPICES methods.

With respect to our chosen study design, setting and sample size, we feel that study design is better mentioned at the start of the methodology section. Study settings are described under ‘study procedures’ as adapted above, and sample size has been added as part of the ‘statistical analysis’ section, based on a power analysis, P.10 line 251-253:

“The sample size is based on outcome parameters (i.e. joint angles) of a previous validity study comparing spastic CP patients with their TD peers, yielding an effect size of 0.91 (20). Based on this effect size, a group of 20 DCP and 20 TD individuals is sufficient.”

With respect to the participants, participants and participant characteristics are described at the start of the methods section, P.6, lines 155-165.

With respect to intervention, the methodology applied in our study agrees more with ‘exposure’, since we did not perform an intervention. The exposure – 3D motion analysis – is described under ‘study procedures’ on P.6, lines 166-205. 

With respect to comparison, we have adapted the wording in the manuscript to emphasize the comparison aspect. Comparison between groups is described as part of the statistical analysis section on P.10, line 246-250:

“Between-group comparison: Sub-goal 4 - Joint angles at PTA and spatio-temporal parameters averaged over repetitions were assessed for normality and compared between groups with an independent t-test/Mann Whitney-U test. Additionally, absolute differences (the difference between the mean of the TD and DCP group) were compared with the between-session standard error to evaluate for which parameters the absolute difference exceeds the standard error.”

With respect to ethics, we have added some words in the manuscripts to emphasize the ethics part, P.6 lines 163-165:

“With respect to ethics, all participants and/or their parents provided written consent prior to participation in accordance with the Declaration of Helsinki. The study was approved by the Ethics committee research UZ / KU Leuven, S-number S62093”.

With respect to statistical analysis, all analyses are described under the section “Statistical analysis”, P.9, lines 221-255.

2. Also please do take a reference from the checklist so that the paper is scientifically sound.

a. Describe the trial design and allocation ratio.

b. What changes were made to methods after trial commencement? Kindly explain with reasons.

c. What were Eligibility criteria for participants?

d. Mention the settings and locations where the data were collected.

e. Define pre-specified primary and secondary outcome measure.

f. How was sample size determined?

g. Who enrolled participants?

h. Who assigned participants to interventions?

i. Statistical methods need more explanation.

Reply: At the beginning of the methods section, we included references with respect to both the Cosmin checklist – referring to reviewer 1 – and the SPICES method – referring to reviewer 2 – P.6 lines 151-153:

“The Cosmin checklist was used for standardisation of reporting of clinimetric properties and we adhered to the SPICES method to ascertain inclusion of all aspects of the methodology (1, 20).”

a. With respect to trial design: We have added a section ‘study design’, discussing the different trial designs, based on the abovementioned comments at the start of the methods section, P.5 lines 147-151:

“Within-session reliability and repeatability were evaluated using the intra-class correlation coefficient and standard error of measurement on the parameters collected within one session. Between-session repeatability was evaluated by using data of the first and second session, and intrinsic and extrinsic variability were explored. All parameters were compared between the TD individuals and individuals with DCP to evaluate between-group differences.”

b. We did not make any changes to the methods after trial commencement

c. With respect to the eligibility criteria: We have changed our wording in the manuscript to improve interpretation, P.6 lines 156-162: 

“Individuals with DCP were eligible to participate if they: were diagnosed with DCP by a paediatric neurologist, were aged between 5-25 years old and were classified as Manual Ability Classification System (MACS) level I-III. Exclusion criteria were: a neurological disorder other than DCP, botulinum toxin injections in the upper limb muscles in the past 6 months and neurological or orthopaedic surgery in the last year before assessment. TD participants were recruited from a peripheral network and eligible to participate if they were aged between 5-25 years old.”

d. With respect to the settings and locations were the data was collected, we have included this in the ‘study procedures’ section on P.6, lines 167-170:

“Every child was evaluated twice on the same day with a minimum of one hour and a maximum of two hours between sessions at the WE-lab for Health, Technology and Management (KU Leuven, campus Bruges) or the Clinical Movement Analysis Laboratory (CMAL, UZ Leuven, Pellenberg) by the same assessors.”

e. With respect to primary and secondary outcome measures, we did not include secondary outcome measures since the joint angles and spatio-temporal parameters are both primary outcome measures and we focused on multiple analyses on the same primary outcome measure.

f. With respect to sample size, we have added sample size determination on P.10, lines 251-253:

“The sample size was based on outcome parameters (i.e. joint angles) of a previous validity study comparing spastic CP patients with their TD peers, yielding an effect size of 0.91 (14). Based on this effect size, a group of 20 DCP and 20 TD individuals was sufficient.”

g. With respect to participant enrolment, multi-disciplinary schools were approached and if they were willing to collaborate, the research team evaluated whether the possible candidates fulfilled the eligibility criteria. 

h. With respect to intervention: This point is not applicable as there was no intervention in the current study.

i. With respect to the statistical methods, the approach we have applied is a rather standard approach for the evaluation of psychometric properties, where we analysed parameters via intra-class correlation coefficients and standard error of measurement. Additionally, we have used the Cosmin standards to report the statistical methods. Our approach is similar to previous studies in CP in both upper limb (19) and gait research (22). We have made some changes in the manuscript to improve readability and interpretation:

“Within-session repeatability: Goal 1 - For both groups, 2, 4, 6, 8 and 10 repetitions from the RF, RGV and RS task were randomly selected for each task for each participant. Subsequently, intra-class correlation coefficient (ICC) values – ICCw(2,1) based on single measures – and standard error of measurement (SEM) were calculated for each number of repetitions for the joint angle at PTA and the spatio-temporal parameters for each functional task. Values of ICC were interpreted as poor (<0.50), moderate (0.50- < 0.75), good (0.75–0.90), and excellent (> 0.90). (2). SEM calculations were based on the square root of the within-group mean square value of the one-way ANOVA (23).

The change in both the ICC and SEM values for the different repetitions (2, 4, 6, 8 or 10) was expressed in percentage (%) of change in comparison with the highest SEM or ICC value for all number of repetitions. The cut-off value for a stable ICC or SEM value was defined as the difference between incrementing repetitions being less than or equal to 10%. The SEM defining the cut-off value will hereafter be referred to as ‘consistency measure’, since we assume that this margin of error defines a consistent performance within one session.

Assessment of variability: Goal 2: To evaluate whether the variability was higher for participants with DCP in comparison with their TD peers, standard deviations for the selected number of repetitions were calculated and compared between groups using an independent t-test/Mann Whitney-U test depending on the data distribution. 

Between-session repeatability: Goal 3 – To evaluate repeatability over time in patients with DCP, it is important to differentiate between internal variability (the difference in consistency only related to the participants’ performance) and external sources of variability (e.g. marker placement and palpation differences). To evaluate internal variability, we compared the consistency measure between session 1 and session 2 for each task. To evaluate external variability we compared the mean of the consistency measures of session 1 and session 2 with the between-session standard error for all joint angles and spatio-temporal parameters. 

Between-group comparison: Goal 4 - Joint angles at PTA and spatio-temporal parameters averaged over repetitions were assessed for normality and compared between groups with an independent t-test/Mann Whitney-U test. Additionally, absolute differences (the difference between the mean of the TD and DCP group) were compared with the between-session standard error to evaluate for which parameters the absolute difference exceeds the standard error.”

References

1. Gagnier JJ, Lai J, Mokkink LB, Terwee CB. COSMIN reporting guideline for studies on measurement properties of patient-reported outcome measures. Quality of life research : an international journal of quality of life aspects of treatment, care and rehabilitation. 2021;30(8):2197-218.

2. Koo TK, Li MY. A Guideline of Selecting and Reporting Intraclass Correlation Coefficients for Reliability Research. J Chiropr Med. 2016;15(2):155-63.

3. Mackey AH, Walt SE, Lobb GA, Stott NS. Reliability of upper and lower limb three-dimensional kinematics in children with hemiplegia. Gait Posture. 2005;22(1):1-9.

4. Butler EE, Ladd AL, Louie SA, Lamont LE, Wong W, Rose J. Three-dimensional kinematics of the upper limb during a Reach and Grasp Cycle for children. Gait Posture. 2010;32(1):72-7.

5. Reid S, Elliott C, Alderson J, Lloyd D, Elliott B. Repeatability of upper limb kinematics for children with and without cerebral palsy. Gait Posture. 2010;32(1):10-7.

6. Jaspers E, Feys H, Bruyninckx H, Cutti A, Harlaar J, Molenaers G, et al. The reliability of upper limb kinematics in children with hemiplegic cerebral palsy. Gait Posture. 2011;33(4):568-75.

7. Brihmat N, Loubinoux I, Castel-Lacanal E, Marque P, Gasq D. Kinematic parameters obtained with the ArmeoSpring for upper-limb assessment after stroke: a reliability and learning effect study for guiding parameter use. Journal of neuroengineering and rehabilitation. 2020;17(1):130.

8. Frykberg G, Grip H, Alt Murphy M. How many trials are needed in kinematic analysis of reach-to-grasp?—A study of the drinking task in persons with stroke and non-disabled controls. Journal of neuroengineering and rehabilitation. 2021;18.

9. Thigpen CA, Gross MT, Karas SG, Garrett WE, Yu B. The repeatability of scapular rotations across three planes of humeral elevation. Res Sports Med. 2005;13(3):181-98.

10. Lempereur M, Brochard S, Leboeuf F, Rémy-Néris O. Validity and reliability of 3D marker based scapular motion analysis: A systematic review. Journal of biomechanics. 2014;47(10):2219-30.

11. McIntyre S, Goldsmith S, Webb A, Ehlinger V, Hollung SJ, McConnell K, et al. Global prevalence of cerebral palsy: A systematic analysis. Developmental Medicine & Child Neurology.n/a(n/a).

12. Himmelmann K, McManus V, Hagberg G, Uvebrant P, Krageloh-Mann I, Cans C. Dyskinetic cerebral palsy in Europe: trends in prevalence and severity. Archives of disease in childhood. 2009;94(12):921-6.

13. Butler EE, Rose J. The pediatric upper limb motion index and a temporal-spatial logistic regression: quantitative analysis of upper limb movement disorders during the Reach & Grasp Cycle. Journal of biomechanics. 2012;45(6):945-51.

14. Jaspers E, Desloovere K, Bruyninckx H, Klingels K, Molenaers G, Aertbelien E, et al. Three-dimensional upper limb movement characteristics in children with hemiplegic cerebral palsy and typically developing children. Res Dev Disabil. 2011;32(6):2283-94.

15. Gordon LM, Keller JL, Stashinko EE, Hoon AH, Bastian AJ. Can spasticity and dystonia be independently measured in cerebral palsy? Pediatric neurology. 2006;35(6):375-81.

16. Sanger TD. Arm trajectories in dyskinetic cerebral palsy have increased random variability. J Child Neurol. 2006;21(7):551-7.

17. Schneiberg S, McKinley P, Gisel E, Sveistrup H, Levin MF. Reliability of kinematic measures of functional reaching in children with cerebral palsy. Dev Med Child Neurol. 2010;52(7):e167-73.

18. Rönnqvist L, Rösblad B. Kinematic analysis of unimanual reaching and grasping movements in children with hemiplegic cerebral palsy. Clinical biomechanics (Bristol, Avon). 2007;22(2):165-75.

19. Jaspers E, Feys H, Bruyninckx H, Harlaar J, Molenaers G, Desloovere K. Upper limb kinematics: development and reliability of a clinical protocol for children. Gait Posture. 2011;33(2):279-85.

20. Booth A. Clear and present questions: formulating questions for evidence based practice. Library Hi Tech. 2006;24(3):355-68.

21. Wu G, van der Helm FC, Veeger HE, Makhsous M, Van Roy P, Anglin C, et al. ISB recommendation on definitions of joint coordinate systems of various joints for the reporting of human joint motion--Part II: shoulder, elbow, wrist and hand. Journal of biomechanics. 2005;38(5):981-92.

22. Verreydt I, Vanderkerkhove I, Stoop E, Peeters N, Van tittelboom V, Van de walle P, et al. Instrumented strength assessment in typically developing children and children with a neural or neuromuscular disorder: a reliability, validity and responsiveness study. Front Physiol. 2022;13.

23. Weir JP. Quantifying test-retest reliability using the intraclass correlation coefficient and the SEM. J Strength Cond Res. 2005;19(1):231-40.

---

## [Decision Letter · Decision Letter 1]

13 Sep 2022

Psychometric properties of upper limb kinematics during functional tasks in children and adolescents with dyskinetic cerebral palsy

PONE-D-22-07917R1

Dear Dr. Vanmechelen,

We’re pleased to inform you that your manuscript has been judged scientifically suitable for publication and will be formally accepted for publication once it meets all outstanding technical requirements.

Kind regards,

Yih-Kuen Jan, PhD

Academic Editor

PLOS ONE

Additional Editor Comments (optional):

Reviewers' comments:

Reviewer's Responses to Questions

**Comments to the Author**

1. If the authors have adequately addressed your comments raised in a previous round of review and you feel that this manuscript is now acceptable for publication, you may indicate that here to bypass the “Comments to the Author” section, enter your conflict of interest statement in the “Confidential to Editor” section, and submit your "Accept" recommendation.

Reviewer #2: All comments have been addressed

2. Is the manuscript technically sound, and do the data support the conclusions?

Reviewer #2: Yes

3. Has the statistical analysis been performed appropriately and rigorously? 

Reviewer #2: Yes

4. Have the authors made all data underlying the findings in their manuscript fully available?

Reviewer #2: Yes

5. Is the manuscript presented in an intelligible fashion and written in standard English?

Reviewer #2: Yes

6. Review Comments to the Author

Reviewer #2: All corrections done by the authors are amended. I have no further comments. I suggest accepting this manuscript in its current version.

7. PLOS authors have the option to publish the peer review history of their article (what does this mean?). If published, this will include your full peer review and any attached files.

Reviewer #2: No

---

## [Editor Report · Acceptance letter]

15 Sep 2022

PONE-D-22-07917R1 

Psychometric properties of upper limb kinematics during functional tasks in children and adolescents with dyskinetic cerebral palsy 

Dear Dr. Vanmechelen:

I'm pleased to inform you that your manuscript has been deemed suitable for publication in PLOS ONE. Congratulations! Your manuscript is now with our production department. 

Kind regards, 

on behalf of

Dr. Yih-Kuen Jan 

Academic Editor

PLOS ONE